# Quercetin 3-O-(6″-O-E-caffeoyl)-β-D-glucopyranoside, a Flavonoid Compound, Promotes Melanogenesis through the Upregulation of MAPKs and Akt/GSK3β/β-Catenin Signaling Pathways

**DOI:** 10.3390/ijms24054780

**Published:** 2023-03-01

**Authors:** Changhai Liu, Mayire Nueraihemaiti, Deng Zang, Salamet Edirs, Guoan Zou, Haji Akber Aisa

**Affiliations:** 1State Key Laboratory Basis of Xinjiang Indigenous Medicinal Plants Resource Utilization, CAS Key Laboratory of Chemistry of Plant Resources in Arid Zone, Xinjiang Technical Institute of Physics and Chemistry, Chinese Academy of Sciences, Urumqi 830011, China; 2University of Chinese Academy of Sciences, Beijing 100049, China

**Keywords:** 3-O-(6″-O-E-caffeoyl)-β-D-glucopyranoside, MAPKs, Akt/GSK-3β/β-catenin, melanogenesis

## Abstract

Quercetin 3-O-(6″-O-E-caffeoyl)-β-D-glucopyranoside is a flavonoid compound produced by various plants with reported antiprotozoal potential against *E. histolytica* and *G. lamblia*; however, its effects on skin pigment regulation have not been studied in detail. In this investigation, we discovered that quercetin 3-O-(6″-O-E-caffeoyl)—D-glucopyranoside (coded as **CC7**) demonstrated a more increased melanogenesis effect in B16 cells. **CC7** exhibited no cytotoxicity or effective stimulating melanin content or intracellular tyrosinase activity. This melanogenic-promoting effect was accompanied by activated expression levels of microphthalmia-associated transcription factor (MITF), a key melanogenic regulatory factor, melanogenic enzymes, and tyrosinase (TYR) and tyrosinase-related protein-1 (TRP-1) and 2 (TRP-2) in the **CC7**-treated cells. Mechanistically, we found that **CC7** exerted melanogenic effects by upregulating the phosphorylation of stress-regulated protein kinase (p38) and c-Jun N-terminal kinase (JNK). Moreover, the **CC7** upregulation of phosphor-protein kinase B (Akt) and Glycogen synthase kinase-3 beta (GSK-3β) increased the content of β-catenin in the cell cytoplasm, and subsequently, it translocated into the nucleus, resulting in melanogenesis. Specific inhibitors of P38, JNK, and Akt validated that **CC7** promotes melanin synthesis and tyrosinase activity by regulating the GSK3β/β-catenin signaling pathways. Our results support that the **CC7** regulation of melanogenesis involves MAPKs and Akt/GSK3β/β-catenin signaling pathways.

## 1. Introduction

Melanogenesis is a complex process in which the pigment melanin is produced in the melanosomes by melanocytes and transported to keratinocytes, which involves a series of enzymatic and multiple signaling pathways. In the melanogenesis pathway, several melanin enzymes are involved; including tyrosinase, TRP-1, and TRP-2. First, L-tyrosine in melanocytes is hydroxylated to L-dihydroxyphenylalanine (L-DOPA). L-DOPA is further oxidized to dopaquinone (DQ), while DQ in the presence of cysteine leads to the formation of 3- or 5-cysteinylDOPAs, which then yield pheomelanin during conversion [1]. MITF is a key regulator of melanogenesis and mainly induces the expression of TYR, TRP-1, and TRP-2 [2]. Several signal transduction pathways that are involved in melanogenesis regulate MITF and TYR expression, including the cAMP/PKA, MAPK, and WNT/β-catenin ones [3]. In has been shown that the phosphorylation of p38 and c-Junction JNK trigger melanogenesis by stabilizing MITF activation [4]. The Wnt/β-catenin pathway plays a crucial role in melanocyte development by inactivatingGSK-3β, leading to the accumulation of β-catenin in the cytoplasm. β-catenin translocates into the nucleus, where it increases the expression of MITF [5]. 

Melanogenesis activity is also influenced by various extrinsic factors, such as ultraviolet radiation and drugs [6]. Abnormally low production of melanogenesis in skin melanocytes results in hypo-pigmentation disorders, such as vitiligo [7]. These hypo-pigmentary skin disorders usually impact the appearance of patients, which may cause psychological implications and a significantly reduced quality of life [8]. Hence, the regulation of melanogenesis is essential for treating hypo-pigmentary disorders. 

Flavonoids are attractive natural compounds that possess a wide range of bioactivities and pharmacological properties by these species, which include anxiolytic [9], antimicrobial [10], neuroprotective [11], anticarcinogenic [12], anti-inflammatory [13], and antiallergic [14] properties. Recently, we found the potential of flavonoids in the therapy of hypopigmentation disorders has been highlighted by many researchers [15,16,17]. However, these different flavonoids exert an inconsistent effect on melanogenesis despite their structural similarity, which exhibits both inhibit melanin and promote melanin, two opposite pharmacological characteristics [18]. Galangin could manage melanin biosynthesis through the competitive inhibition of tyrosinase [19], but many other flavonoids promote melanin production by regulating MITF gene expression via different signaling pathways [20].

Quercetin 3-O-(6″-O-E-caffeoyl)-β-D-glucopyranoside (**CC7**) is a flavonoid compound produced by various plants with reported antiprotozoal potential against *E. histolytica* and *G. lamblia* [21]; however, its effects on melanogenesis regulation have not been studied in detail until now. Hence, in this study, we investigated Quercetin 3-O-(6″-O-E-caffeoyl)-β-D-glucopyranoside (encoded as **CC7**) (Figure 1) melanogenesis effects and its mechanism in vitro using B16 cells. Our results indicated that the compound Quercetin 3-O-(6″-O-E-caffeoyl)-β-D-glucopyranoside exhibited the most potential to boost melanogenesis activity. It is essential to developing **CC7** as a novel pigment disorder ingredient. However, further clinical or animal model research must be carried out.

## 2. Results

### 2.1. Toxicity Test of the **CC7** on B16 Cells

The CCK-8 assay was used to assess the safety of **CC7** from 1 to 50 μM for the cell viability of the B16 melanoma cells. **CC7**-treated B16 revealed that cell viability was not affected from 1 μM to 50 μM. Figure 1B confirms that **CC7** did not show any toxicity effects on increase in the cell number of the B16 cells (Figure 1B). Further testing shows that the value of IC50 of **CC7** against B16 was 0.464 mM.

### 2.2. Melanogenic Effect of the **CC7** on B16 Cells and PIG3V Melanocytes

To verify the melanogenesis effect of **CC7**, we measured the intracellular melanin contents and tyrosinase activity in the B16 cells. Based on the cell viability results, the further melanogenic experiment on **CC7** focused on non-cytotoxic concentrations of 1 μM to 50 μM. Our results revealed that the application of 1 μM to 50 μM **CC7** dose-dependently promotes the intracellular production of melanin and tyrosinase activity in B16 cells (Figure 2A,B). We used 8-MOP as a positive control for the melanogenic activity test. Next, we further investigated the **CC7** melanogenic effect on the human epidermal melanocytes cell line, PIG3V. As shown in Figure 1D, **CC7** significantly increased the cellular melanin contents at concentrations of 50 µM (*p* < 0.0001).

### 2.3. **CC7** Promotes the Protein Expression Levels of MITF and TRPs in B16 Cells 

To determine the hypotheses that **CC7** positively regulates the melanin content and tyrosinase enzymatic activity is affected by the protein expression levels of melanogenic enzymes, tyrosinase (TYR), tyrosinase-related protein 1 and 2 (TRP1 and TRP2, respectively), and of these enzymes’ transcription factor, microphthalmia-associated transcription factor (MITF), we next performed a Western blotting analysis. The results showed that **CC7** dose-dependently enhanced the protein expression levels of MITF, tyrosinase, TRP-1, and TRP-2 in B16 cells for 48 h compared to those of the untreated control groups (Figure 3A,B). These results suggest that **CC7** promotes melanin synthesis by the MITF-mediated upregulation of TYR, TRP-1, and TRP-2 at a cellular level.

### 2.4. **CC7** Promotes the Protein Expression Levels of p38MAPKs and JNK in B16 Cells

To further investigate the molecular mechanisms of **CC7 to** induce melanin production, the B16 cells were treated with **CC7** at different concentrations (1, 10, and 50 μM) or NC for 48 h. The phosphorylated expression levels, and the total amount of ERK, P38, and JNK were examined by Western blotting. The results showed that the expression level of phospho-p38 and phospho-JNK were upregulated in response to **CC7** in the B16 cells. However, as presented in Figure 4A, the expression level of ERK was not increased by **CC7**. 

### 2.5. **CC7** Enhances the Melanogenesis Not Affected by MAPKs in B16 Cells

To further confirm the above results, we used the PD98059 (a specific ERK inhibitor), SP600125 (a JNK inhibitor), and SB203580 (a p38 inhibitor) co-treated with **CC7** on B16 cells incubated for 48 h to obtain measurements of the melanin content and tyrosinase activity. Figure 5A,B confirms that the melanogenic activity induced by **CC7** was reduced by SP600125 and SB203580. However, PD98059 did not inhibit the reduction of the melanin content and tyrosinase activity by **CC7**. These results suggest that **CC7** may cause melanogenesis via the p38 MAPK and JNK signal pathways, but not ERK MAPK.

### 2.6. **CC7** Promotes the Protein Expression Levels of Akt, GSK3β, and β-Catenin in B16 Cells

We further determined whether the Akt/GSK3β/β-catenin was involved in the melanogenic activity of **CC7** by Western blotting. As shown in Figure 6A, the expression levels of phosphor-Akt, phosphor-GSK3β, phosphor-β-catenin at ser 675, and β-catenin were upregulated in response to **CC7** in the B16 cells. In contrast, phosphor-β-catenin at the ser 33,37,41 levels decreased in a dose-dependent manner.

### 2.7. Melanogenic Effect of **CC7** Involves Upregulating the Akt/GSK3β/β-Catenin in B16 Cells

Moreover, we analyzed β-catenin in cytoplasmic and nuclear fractions in the B16 cells. After the treatment with **CC7** for 48 h, the protein level of β-catenin in the nuclear fraction was significantly increased (Figure 7A,B). Then, we also further confirmed the role of the Akt/GSK3β/β-catenin pathways in **CC7**-induced melanogenesis by applying the Akt inhibitor co-treated with **CC7** to the B16 cells incubated for 48 h for measurements of the melanin content and tyrosinase activity. As shown in Figure 7C, we found that an Akt inhibitor reduced the melanogenic activity induced by **CC7**. 

## 3. Discussion

The abnormally low regulation of melanogenesis leading to hypo-pigmentation is found in human diseases such as vitiligo. Some natural products can alter pigment production and the expression of key melanogenesis transcription factors and enzymes [22]. Flavonoids are available in a wide range of plant components. Some flavonoids have been developed into therapeutic products due to their beneficial bioactivities and low toxicity. Many flavonoids have been reported to up and downregulate the effects on melanogenesis [23,24,25]. Huang et al. [26] reported that exposure of B16F10 cells to 100 μM naringenin (a naturally occurring citrus flavanone) resulted in upregulated melanogenesis by promoting the expression level of MITF and the tyrosinase family gene. Moreover, the activity of phosphatidyl-inositol 3-kinase (PI3K) and its downstream Akt protein was upregulated by naringenin, and also, an increase in β-catenin was observed, as well as the phosphor-GSK3β. In this study, we found a flavonoid compound, Quercetin 3-O-(6″-O-E-caffeoyl)-β-D-glucopyranoside (encoded as **CC7**), that exhibited highly induced melanogenesis in B16 melanoma cells, and further, we used molecular approaches to understand the regulating pathways involved in the melanogenic effect. We first evaluated the cytotoxicity and determined that the safe concentration range of **CC7** was 1 μM to 50 μM by an CCK-8 assay. Thus, we used a 1–50 μM concentration of **CC7** for the further melanogenic experiment. Next, we found that **CC7** significantly promotes cellular melanin contents and tyrosinase activity in B16 cells. 

Several major enzymes are involved in the melanogenesis of melanocytes. TYR is the rate-limiting enzyme in melanogenesis [27]. TRP-1 and TRP-2 catalyze the dopachrome conversion to melanin, which consequently induces melanin synthesis [28]. These enzyme expression levels are regulated by MITF, a major regulator of melanogenesis [29]. In this study, **CC7** significantly increased the expression of TYR, TRP-1, TRP-2, and MITF. These results suggest that **CC7** promotes the melanin content and tyrosinase activity through the upregulation of MITF and its downstream three melanogenesis-related enzymes in B16 cells. 

Previous research has reported that MAPK signaling pathways (ERK, JNK, and p38) participate in the activation of MITF, which then induces melanin synthesis [30,31,32]. It has been shown that the phosphorylation of p38 and c-Junction JNK trigger melanogenesis by stabilizing MITF activation [33]. Therefore, in this study, we investigated whether **CC7** induces JNK and p38 MAPK or ERK repression in B16 cells. In our results, the expression level of phosphor-p38 and phospho-JNK were upregulated in response to **CC7** in the B16 cells. However, in **CC7,** no effect was observed in the protein expression levels of the phosphorylated of ERK. In addition, that melanogenic activity induced by **CC7** was reduced by SP600125 and SB203580. However, compared to the treatment with **CC7** alone, in a co-treatment with PD98059 (a specific ERK inhibitor), **CC7** did not change the melanin contents and tyrosinase activity significantly. These results indicate that **CC7** can enhance the phosphorylation of p38 and c-Junction JNK trigger melanogenesis by stabilizing MITF activation. Thus, the p38 and JNK MAPKs pathways are involved in **CC7**-induced melanogenesis, but not ERK.

The relationship between Akt/GSK-3β/β-catenin signaling and melanogenesis has been demonstrated by many studies [34]. GSK-3β is a negative regulator protein in the Wnt/β-catenin signaling pathway and is phosphorylated by several kinases, including Akt [35]. Activated GSK-3β induces the phosphorylation of β-catenin, after which the β-catenin is ubiquitination and degraded [36]. When phosphorylation of GSK-3β at Ser9 inhibits the activity of GSK-3β and further inhibits the degradation of β-catenin, β-catenin can accumulate in the cytoplasm and translocate into the nucleus, and it promotes MITF transcription, and eventually, it promotes the biosynthesis of melanin [37,38,39]. Our results showed that the **CC7** treatment upregulated the phosphorylation of Akt and GSK-3β, which lead to the accumulation of β-catenin in the cytoplasm. Consistent with this point, our results showed that **CC7** promotes the accumulation of β-catenin in the cytoplasm, and the protein level of β-catenin in the nuclear fraction was significantly increased. B16 cells were co-treated with CC7 and incubated for 48 h to measure the melanin content and tyrosinase activity. We found that the Akt inhibitor reduced the melanogenic activity induced by **CC7**. Thus, the melanin production mediated by the compound **CC7** was probably triggered through Akt/GSK-3β/β-catenin pathways, which is consistent with reports revealing that the activation of GSK3/β-catenin influenced melanin production in B16 melanoma cells. 

## 4. Materials and Methods

### 4.1. Drug Preparation and Materials

**CC7** was obtained from the State Key Laboratory Basis of Xinjiang Indigenous Medicinal Plants Resource Utilization. As a yellow powder, the molecular formula was determined as C_30_H_26_O_15_ based on the ion peak by HR-ESI-MS at *m*/*z* [M+H]+ = 627.1344, NMR (Appendix A Table A1 and Appendix B Figure A1). The NMR spectra indicated that **CC7** has a structure consistent with that which has been previously reported [21]. The purity of the **CC7** was determined by HPLC using the MeOH and H_2_O solvent system, and the purity was 98% (Appendix B). L-3-(3,4-Dihydroxyphenyl) alanine (L-DOPA) (CAS:59-92-7) and 8-Methoxypsoralen (8-MOP) (CAS:298-81-7) were purchased from Sigma Aldrich (Milan, Italy). Antibodies for tyrosinase (C-19), TRP-1 (H-90), and TRP-2 (H-150) were purchased from Santa Cruz Biotechnology (Santa Cruz, Dallas, TX, USA). MITF (D5G7V), CREB (86B10), p-CREB (Ser133) (1B6), p38 (L53F8), p-p38 (Thr180/Tyr182) (28B10), SAPK/JNK (#9252s), p-JNK (Thr183/Tyr185), ERK (L34F12), and-ERK (Thr202/Tyr204) (E10) were purchased from Cell Signaling Technology (Danvers, MA, USA). Antibodies for lamin-B were obtained from Absin Bioscience Inc. (Shanghai, China) β-actin (BA2305), Goat anti-rabbit (BA1054), goat anti-mouse (BA1050), and rabbit anti-goat (BA1060) antibodies were obtained from BOSTER Biological Technology (Wuhan, China). AKT inhibitor VIII (CAS:612847-09-3), SP600125 (CAS:129-56-6), SB203580 (CAS:152121-47-6) and the Nuclear and Cytoplasmic extraction kit were obtained from Beyotime Technology (Shanghai, China).

### 4.2. Cell Culture

Murine melanoma B16 cell line (Cat# TCM2) which were cultured Dulbecco’s modified eagle medium (Gibco Life Technologies, Waltham, MA, USA) supplemented with 10% heat-inactivated fetal bovine serum (Gibco Life Technologies, Waltham, MA, USA), 100 Units/mL penicillin G, and 100 µg/mL streptomycin (Gibco BRL, Grand Island, NY, USA) at 37 °C in a 5% CO_2_ incubator (Thermo Fisher Scientific, Waltham, MA, USA). PIG3V melanocytes were provided by Dr. Caroline Le Poole (Loyola University Chicago, Maywood, IL, USA) and maintained in Medium 254 with Human Melanocyte Growth Supplement (Gibco Life Technologies, Waltham, MA, USA), 5% FBS, penicillin G (100 U/mL), and streptomycin (100 µg/mL) at 37 °C in the presence of 5% CO_2_.

### 4.3. Cell Viability Assay

The viability of the B16 cells was evaluated using a CCK-8 assay as previously described [40]. Briefly, the B16 cells were seeded at a density of 5 × 10^3^ cells per well in 96-well and incubated with **CC7** at a concentration of 1–50 µM for 24 h. Then, the culture medium was replaced with the CCK-8 (Absin, Shanghai, China) solution (10 µL). Then, the cells were further incubated for 2 h at 37 °C, and the absorbance was read at 450 nm using a microplate reader Spectra Max M5 (Molecular Devices Company, San Diego, CA, USA). An equal volume of cells without treatment was used as a blank control. All the experiments were repeated three times. 

### 4.4. Tyrosinase Activity

An intracellular tyrosinase activity test was performed by measuring the rate of L-3, 4dihydroxyphenylalanine (L-DOPA) oxidase activity [41,42]. Briefly, the B16 cells were seeded into six-well plates at 3.5 × 10^5^ cells per well and treated with **CC7** for 24 h at 37 °C. The cells were then washed with cold PBS and lysed in a PBS buffer containing 1% TritonX-100 + 1% sodium deoxycholate. The cell lysates were centrifuged at 12,000× *g* for 20 min, and 90 µL of this supernatant and 10 µL of 10 mM L-DOPA solution were mixed and plated in 96-well plates and incubated at 37 °C for 30 min. The optical densities were measured at 490 nm using a microplate reader.

### 4.5. Melanin Contents Measurement

The melanin contents were measured as previously described [43]. Briefly, the B16 cells and PIG3V were seeded in six-well plates (2 × 10^5^ cells/well), respectively, and incubated at various (1–50 µM) concentrations of **CC7** for 48 h at 37 °C. After this, the cells were lysed by RIPA buffer and centrifuged at 12,000× *g* for 20 min. The total protein content was determines using the BCA kit assay (PP02) (Biomed, Beijing, China). Then, 190 µL 1 mM NaOH was added to the cell pellet at 80 °C for 1 h. Additionally, the melanin content was measured at 405 nm and normalized to the total protein content.

### 4.6. Western Blot Analysis

The protein sample of the B16 cells was prepared as mentioned in the previous section above. Nucleus and cytoplasm protein separation experiments were performed using the Nucleoplasmic and Cytoplasmic Protein Extraction Kit (Beyotime Technology, Shanghai, China) following the manufacturer’s protocol. All the proteins samples were separated by 10% SDS-PAGE, the proteins were transferred to PVDF membrane, and then the membrane was blocked with 5% skim milk in TBST at 20 °C for 1 h, then exposed overnight at 4 °C with specific appropriate antibodies. Following this, they were washed with TBST three times, and then incubation with the second antibody at 20 °C for 1 h, and after washing them three times with TBST, the protein band signals were detected using an ECL (enhanced chemiluminescence) Western blotting detection kit and quantified using a Chemi Doc MP Imaging system (Bio-Rad Laboratories, Inc., Hercules, CA, USA). All the experiments were performed three times.

### 4.7. Statistical Analysis

All the results are expressed as mean ± SD; the statistical analysis was performed with one-way ANOVA, followed by Tukey’s multiple comparisons tests. The statistical analysis was performed using GraphPad Prism 9 (La Jolla, CA, USA). *p*-values < 0.05 were considered to be statistically significant.

## 5. Conclusions 

In conclusion, Quercetin 3-O-(6″-O-E-caffeoyl)-β-D-glucopyranoside exhibited the best potential to boost the melanogenesis activity by upregulating melanin synthesis and tyrosinase in vitro in the B16 cells. This effect was observed via the activation of the MAPKs and Akt/GSK-3β/β-catenin signal pathways (graphical abstract). This is essential to the possibility of developing Quercetin 3-O-(6″-O-E-caffeoyl)-β-D-glucopyranoside as a novel pigment disorder ingredient. However, further clinical or animal model research must be carried out. 

## Figures and Tables

**Figure 1 ijms-24-04780-f001:**
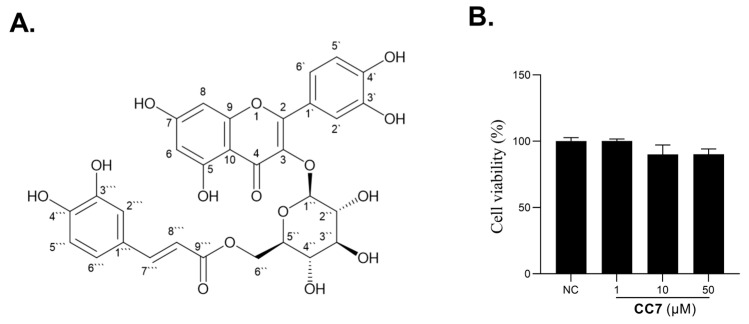
Toxicity test of the **CC7** on B16 cells (**A**) Chemical structure of Quercetin 3-O-(6″-O-E-caffeoyl)-β-D-glucopyranoside (**CC7**). (**B**) Effect of **CC7** on B16 cell viability was measured using a CCK-8 assay kit. All results are represented as means ± SD of three independent experiments. Statistical chart expressed as a percentage relative to the negative control group.

**Figure 2 ijms-24-04780-f002:**
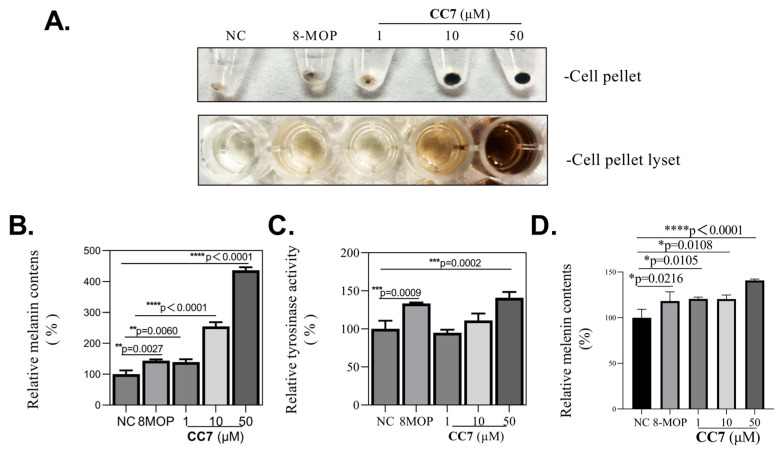
Melanogenic efficacy of **CC7** in B16 melanocytes. (**A**) The effect of melanogenesis by **CC7** and the released melanin pellet was examined. (**B**) Intracellular melanin content and (**C**) tyrosinase activity was determined by measuring the absorbance of B16 cell lysates at 405 nm and 490 nm, respectively. Additionally, it normalizes the obtained values to the protein quantity using a BCA assay kit. Fifty μM 8-MOP was used as the positive control. (**D**) Effect of **CC7** on melanin content of PIG3V melanocytes treated with different concentrations (1, 10, and 50 µM) of **CC7** for 48 h. All results are represented as means ± SD of three independent experiments. Statistical chart expressed as a percentage relative to the negative control group. Statistical analysis was performed with one-way ANOVA, followed by Tukey’s multiple comparisons tests. * *p* < 0.05, ** *p* < 0.01, and *** *p* < 0.001, **** *p* < 0.0001 compared with NC (negative control) group.

**Figure 3 ijms-24-04780-f003:**
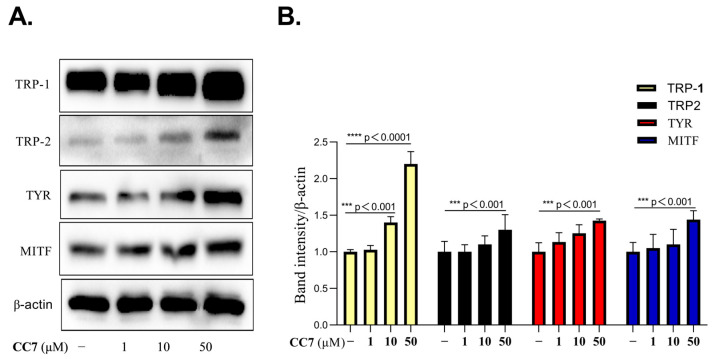
The effects of **CC7** on the protein expression of MITF and TRPs in B16 cells. (**A**) Western blotting results of B16 cells were treated with **CC7** in a safe concentration range (1, 10, and 50 μM) for 48 h; protein levels of MITF, TYR, TRP-1, and TRP-2. All proteins were analyzed relative to β-actin expression. (**B**) The quantification of western bands, and results are presented as the mean ± SD from three independent measurements using Image J software. Statistical analysis was performed with one-way ANOVA, followed by Tukey’s multiple comparisons tests. *** *p* < 0.001, **** *p* < 0.0001 compared with NC (negative control) group.

**Figure 4 ijms-24-04780-f004:**
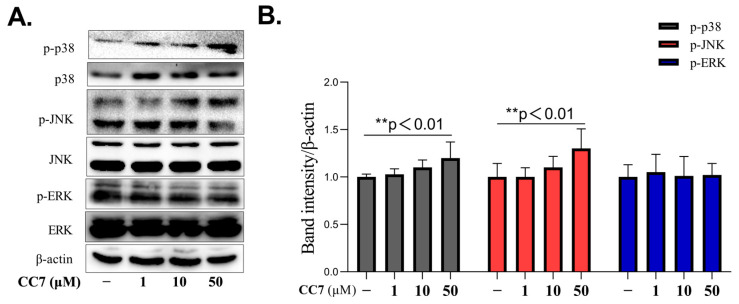
The effects of **CC7** on the protein expression of MAPKs in B16 cells. (**A**) Western blotting results of protein levels of p-ERK, p-JNK, and p-P38 in B16 cells treated with **CC7** in a safe concentration range (1, 10, and 50 μM) for 48 h. All proteins were analyzed relative toβ-actin expression. (**B**) The quantification of western bands, and results are presented as the mean ± SD from three independent measurements using Image J software. Statistical analysis was performed with one-way ANOVA, followed by Tukey’s multiple comparisons tests. ** *p* < 0.01 compared with NC (negative control) group.

**Figure 5 ijms-24-04780-f005:**
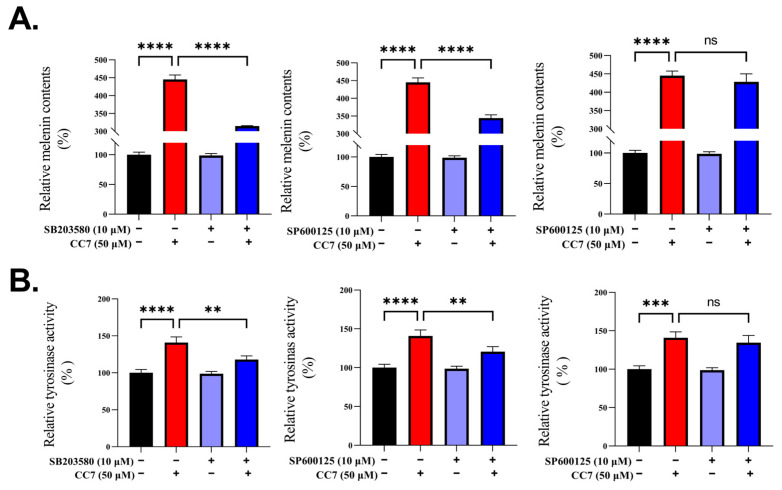
**CC7** enhances the melanogenesis not affected by MAPKs in B16 cells. Effects of **CC7** and in the presence or absence of PD98059, SP600125, and SB203580 on melanin content (%) (**A**) and tyrosinase activity (%) (**B**) in B16 cells after 48 h. Each value is presented as the mean ± SD. ** *p* < 0.01, *** *p* < 0.001 and **** *p* < 0.0001 compared with NC (negative control) group. ns: no statistical significance.

**Figure 6 ijms-24-04780-f006:**
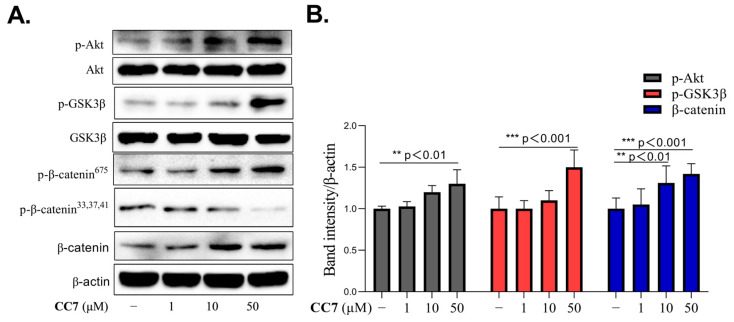
The effects of **CC7** on Akt/GSK3β/β-catenin signal pathway in B16 cells. (**A**) Western blotting results of protein levels of *p*-Akt, p-GSK3β, p-β-catenin, and β-catenin in B16 cells treated with **CC7** at different concentrations (1, 10, and 50 μM) for 48 h. All proteins were analyzed relative to β-actin expression. (**B**) The quantification of western bands and results are presented as the mean ± SD from three independent measurements using Image J software. ** *p* < 0.01, *** *p* < 0.001 compared with NC (negative control) group.

**Figure 7 ijms-24-04780-f007:**
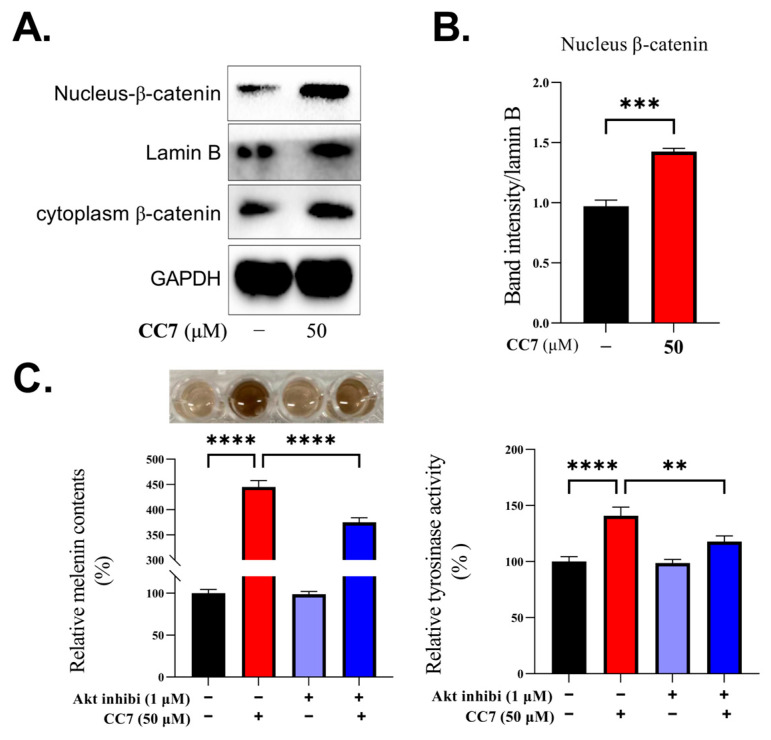
The effects of **CC7** on Akt/GSK3β/β-catenin signal pathway in B16 cells. (**A**) Western blotting results of protein expression levels of β-catenin in cytoplasmic and nuclear fractions in B16 cells after treatment with **CC7** at 50 μM. (**B**) The quantification of western bands, and results are presented as the mean ± SD from three independent measurements using Image J software. (**C**) Effects of **CC7** and in the presence or absence of Akt inhibitor on melanin content (%) and tyrosinase activity (%) in B16 cells after 48 h. red bar was the presence of **CC7** only, purple bar was the presence of Akt inhibitor, blue bar was the cotreatment with Akt inhibitor and **CC7**. The statistical analysis was performed with one-way ANOVA, followed by Tukey’s multiple comparisons tests. ** *p* < 0.01, *** *p* < 0.001 and **** *p* < 0.0001 compared with NC (negative control) group.

## Data Availability

Not applicable.

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
