# Peer review of "Quercetin 3-O-(6″-O-E-caffeoyl)-β-D-glucopyranoside, a Flavonoid Compound, Promotes Melanogenesis through the Upregulation of MAPKs and Akt/GSK3β/β-Catenin Signaling Pathways"

_ijms, 2023, doi:10.3390/ijms24054780_

Round 1

Reviewer 1 Report

I read the paper interestingly and it was a good time. Supplementary explanations and experiments are likely to be needed in some areas, and opinions are given to the author in those areas.

1. In the case of MITF Gene, the high expression level time is generally less than 2 hours. Is there a special reason why you watched MITF in 48 hours? It is recommended to describe the time dependent expression level of MITF, TRP-1, TRP-2, and Tyrosinase by CC7 product.

2. Since melanogenesis occurs frequently in B16 cells, it seems that research on melanogenesis activation of CC7 has been found.
It seems that there is a novelty as a potential treatment for CC7 only when we check if melanogenesis occurs using Vitiligo Melanocyte Cell Line. If possible, it is recommended to describe the result of producing melanin contents using the Vitiligo Melanocyte Cell Line (PIG3V).

3. In the research results conducted in Figure 7, it is expected to be more visualized by showing the generated photo of melanin as shown in Figure 2.

4. Since melanocyte and Keratinocyte are directly connected, if there are research results such as stress mechanism in keratinocyte, we can appeal to the melanocyte-specific signal of CC7 product.

Finally, it is recommended to write a summary figure that can organize the research results and register them as a figure.

I hope the details were helpful to the author and hope for good results.

Reviewer 2 Report

Please refer to the attached file

Round 2

Reviewer 2 Report

The authors provide a satisfactory response to each query. But still, need to improve the figure. Most of the figure values or name is not clear in the present form. So I suggest increasing the size of the figure value/name for clear visualization. 

Author Response

The authors provide a satisfactory response to each query. But still, need to improve the figure. Most of the figure values or name is not clear in the present form. So I suggest increasing the size of the figure value/name for clear visualization.

Response: Thank you for your advice. We redraw these figures with full clear visualization in the present revised manuscript.